Beneficial effects of upgrading to His-Purkinje system pacing in patients with pacing-induced cardiomyopathy: a systematic review and meta-analysis

Tang Nian
Chen Xiaoxiao
Li Hongfei
Zhang Denghong 64518190@qq.com
Geriatric Diseases Institute of Chengdu/Cancer Prevention and Treatment Institute of Chengdu, Department of Cardiology, Chengdu Fifth People’s Hospital (The Second Clinical Medical College, Affiliated Fifth People’s Hospital of Chengdu University of Traditional Chinese Medicine) , Chengdu , China
Shi Huashan
Electronic publication date: 2023 Oct 11
Publication date: 2023
Volume: 11
Electronic Location ID: e16268
Received 2023 Jun 28; Accepted 2023 Sep 19
Copyright: © 2023 Tang et al.
Copyright year: 2023
Copyright holder: Tang et al.
License: This is an open access article distributed under the terms of the Creative Commons Attribution License, which permits unrestricted use, distribution, reproduction and adaptation in any medium and for any purpose provided that it is properly attributed. For attribution, the original author(s), title, publication source (PeerJ) and either DOI or URL of the article must be cited.
License URL: https://creativecommons.org/licenses/by/4.0/

Keywords: His-purkinje system packing, Left bundle branch pacing, Pacing-induced cardiomyopathy, Systematic review, Meta-analysis

Funding: National Natural Science Foundation of China in 2020 82004066 This study was approved by the National Natural Science Foundation of China in 2020—Research on the effect and mechanism of total flavone of Dracocephalum heterophyllum on coronary microcirculation disorder in hypertensive rats based on autophagy regulation pathway (Project No: 82004066). The funders had no role in study design, data collection and analysis, decision to publish, or preparation of the manuscript.

==============================
Background

The purpose of this study was to evaluate the effectiveness of His-Purkinje system pacing (HPSP) in the management of patients with pace-induced cardiomyopathy (PICM).

Methods

PubMed, Embase, Web of Science, and the Cochrane Library were searched comprehensively to collect related studies published from the inception of databases to June 1, 2022. R 4.04 software, including the Metafor package, matrix package, and the Meta package, was utilized to conduct the singe-arm meta-analysis. The methodology index for non-randomized studies (MINORS) was used to assess the methodological quality of the included studies.

Results

A total of seven studies were included, involving 164 PICM patients. The meta-analysis showed that HPSP ameliorated the left ventricular ejection fraction (LVEF) by 13.41% (95% CI [11.21–15.61]), improved the New York Heart Association (NYHA) classification by 1.02 (95% CI [−1.41 to −0.63]), and shortened the QRS duration (QRSd) by 60.85 ms (95% CI [−63.94 to −57.75]), resulting in improved cardiac functions in PICM patients. Besides, HPSP reversed the ventricular remodeling, with a 32.46 ml (95% CI [−53.18 to −11.75]) decrease in left ventricular end systolic volume (LVESV) and a 5.93 mm (95% CI [−7.68 to −4.19]) decrease in left ventricular end-diastolic dimension (LVEDD). HPSP also showed stable electrical parameters of pacemakers, with a 0.07 V (95% CI [0.01–0.13]) increase in pacing threshold, a 0.02 mV (95% CI [−0.85 to 0.90]) increase in sensed R-wave amplitude, and a 31.12 Ω reduction in impedance (95% CI [−69.62 to 7.39]). Compared with LBBP, HBP improved LVEF by 13.28% (95% CI [−11.64 to 14.92]) vs 14.43% (95% CI [−13.01 to 15.85]), ameliorated NHYA classification by 1.18 (95% CI [−1.97 to −0.39]) vs 0.95 (95% CI [−1.33 to −0.58]), shortened QRSd by 63.16 ms (95% CI [−67.00 to −59.32]) vs 57.98 ms (95% CI [−62.52 to −53.25]), and decreased LVEDD by 4.12 mm (95% CI [−5.79 to −2.45]) vs 6.26 mm (95% CI [−62.52 to −53.25]). The electrical parameters of the pacemaker were stable in both groups.

Conclusions

This meta-analysis showed that HPSP could significantly improve cardiac function, promote reverse remodeling, and provide stable electrical parameters of pacemakers for PICM patients.

Introduction

Right ventricular pacing (RVP) induces nonphysiological atrioventricular activation, which has been found to have harmful effects on hemodynamics (Sweeney et al., 2003). For instance, RVP-induced cardiomyopathy (PICM) is a common cause of cardiac dysfunction. Specifically, RVP causes ventricular electromechanical dyssynchrony, which not only adversely affects ventricular structure and function but also increases the risk of heart failure and re-hospitalization (Kiehl et al., 2016; Cho et al., 2019). Cardiac resynchronization therapy (CRT) has been reported to improve PICM in some studies (Gwag et al., 2017; Khurshid et al., 2018; Brignole et al., 2013). CRT is commonly used in patients with heart failure to reverse ventricular remodeling by restoring more synchronized contraction of both ventricles. However, the incidence of PICM remains relatively high. Additionally, CRT is not always feasible, and plenty of patients do not respond to CRT (Funck et al., 2014). In contrast, His-Purkinje system pacing (HPSP) is considered as the most physiologic cardiac pacing mode, including His bundle pacing (HBP) and left bundle branch pacing (LBBP). In 2000, HBP was first introduced in clinical practice by Deshmukh et al. (2000). This mode captures bundle or para-hisian ventricular tissue through the electrical stimulation produced by the pacemaker and ensures normal ventricular electromechanical synchrony via physiological electrical conduction (Ellenbogen & Padala, 2018). HBP has been reported to outperform conventional CRT in ameliorating ventricular electromechanical dyssynchrony, leading to improved cardiac function (Vijayaraman et al., 2019b). LBBP, as an emerging pacing method based on HBP, has attracted great attention in the field of cardiac pacing since it was first reported by Huang et al. (2017). Compared with HBP, LBBP is relatively simple with a higher success rate and may produce better left ventricle electromechanical synchrony (Qian et al., 2020).

HPSP has been reported as an effective cardiac resynchronization therapy (Brignole et al., 2013), but the efficacy of upgrading RVP to HPSP in the management of PICM remains to be elucidated. Besides, whether HBP and LBBP differ in safety and efficacy is unknown. Currently, relevant studies are mostly non-randomized trials with small sample sizes. Despite the publication of more large-scale clinical trials, whether HPSP is effective for PICM patients remains highly debated and inconclusive. Therefore, this systematic review and meta-analysis was conducted to determine the effectiveness of HPSP in PICM patients, and the safety and efficacy of HBP and LBBP were also compared.

Materials and Methods

This study was reported in accordance with the Preferred Reporting Items for Systematic Reviews and Meta-Analyses (Page et al., 2021). In addition, this study protocol has been registered in PROSPERO (ID: CRD42022339804). The approval of the local ethics committee is not required.

This study was conducted according to the following steps: research direction determination-literature retrieval-literature screening-data extraction-analysis of results-writing. PROSPERO is a prospective registration platform that allows registration before data extraction. For prospective registrations, it is possible and reasonable to make appropriate minor adjustments according to actual scenarios in the course of actual research. To minimize the differences between the final manuscript and the protocol registered on the PROSPERO, we registered the study protocol after completing the literature screening.

Search strategy

PubMed, Embase, Web of Science, and the Cochrane Library were comprehensively searched for all related studies published from the database inception to June 1, 2022. The search terms included His-Purkinje system pacing, HPSP, His-Purkinje conduction system pacing, left bundle branch, LBBP, left bundle branch area pacing, LBBAP, left ventricular septal pacing, His-bundle pacing, HBP, His bundle pacing, Hisian pacing, para-Hisian pacing, with the use of Boolean logic operator “AND” and “OR”.

Inclusion and exclusion criteria

The following observational studies (prospective/retrospective) were included: (1) comparing left ventricular ejection fraction (LVEF), New York Heart Association (NYHA) classification, heart Doppler ultrasound, electrical parameters of pacemaker, and follow-up time before and after HPSP upgrading; (2) reporting the specific definition of PICM if PICM patients were enrolled; (3) follow-up duration > 3 months. The following studies were excluded: (1) upgrading to a pacing method other than the His-Purkinje system pacing (2) lacking basic data, such as LVEF, and follow-up time.

Study screening

The retrieved studies were imported into EndnoteX9. After removing duplications, we checked the titles and abstracts to exclude irrelevant studies. Then according to a full-text review, we selected the eligible studies for the meta-analysis. Two reviewers (N. T. and X. C.) independently screened studies and cross-checked their results. If there were disagreements, a third review (H. L.) was consulted.

Data extraction and quality assessment

Two reviewers (N. T. and D. Z.) independently extracted data, and any disagreement was solved through discussion. The following data were extracted: study characteristics (the author, published year, study type, sample size), demographics of patients (male percentage, mean age), clinical characteristics (diagnosis of patients, success rate and good response, RVP continuation time, follow-up time after upgrading), outcomes (LVEF before and after HPSP upgrading or ∆LVEF (LVEF after upgrading minus LVEF before upgrading); NYHA classification before and after HPSP upgrading or ∆NYHA (NYHA after upgrading minus NYHA before upgrading); echocardiographic parameters before and after HPSP upgrading, including LVESV, LVEDD before and after HPSP upgrading, or ∆LVESV (LVESV after upgrading minus LVESV before upgrading), ∆LVEDD (LVEDD after upgrading minus LVEDD before upgrading)); electrical parameters of pacemaker including pacing threshold, sensed R-wave amplitude, impedance before and after HPSP upgrading. Two reviews (N. T. and D. Z.) independently assessed the risk of bias in the included studies, and any disagreement was solved through discussion. Quality assessment was conducted using the methodology index for non-randomized studies (MINORS). The quality assessment tool included 12 items; each item scored from 0 to 2. The first eight items were for the studies without control groups, with a maximum total score of 16. All 12 items were designed for the studies with control groups, with a maximum total score of 24. A score of 0 indicates no report; A score of 1 indicates that insufficient information was reported. A score of 2 indicates that sufficient information was reported.

Outcomes

In this systematic review, the outcomes included LVEF, QRS duration (QRSd), NYHA classification, echocardiography parameters (LVESV, LVEDD), and electrical parameters of pacemaker (pacing threshold, perception, impedance). For continuous outcomes, we used their changes from baseline (before HPSP upgrading) for meta-analysis.

Statistical analysis

The singe-arm meta-analysis was performed using R 4.04 software (R Core Team, 2021), including the Metafor package, matrix package, and Meta package. We calculated the changes in outcome measures from baseline and 95% confidence interval by using a fixed- or random-effects model. Heterogeneity was measured by using the I2 statistic. Q test was used to test the significance of heterogeneity. If I2 < 50%, the fixed-effects model was used. If I2 > 50%, the random-effects model was used. Subgroup analysis was conducted by follow-up time (6 months vs 12 months or longer) and pacing model (HBP vs LBBP).

Results

Study selection

The study selection process is illustrated in Fig. 1. A total of 311 publications were initially identified. A total of 111 duplicate articles were removed, and then 179 publications were excluded after screening titles and then abstracts. The full texts of 21 publications were downloaded and assessed for eligibility. Of the 21 full-text publications, five studies reported no outcomes of interest, and the full text of one study is unavailable. Finally, seven studies were eligible and included in this analysis. All seven eligible studies were observational studies (prospective/retrospective).

Figure 1 Flow diagram of the study selection process.

Study characteristics and quality assessment

A total of seven studies were included, involving 164 PICM patients (seven patients on HBP, seven patients on LBBP), of whom 50–82% were male. The mean age ranged from 69.69 ± 13.75 to 77 ± 10 years. Among the included studies, six studies (Shan et al., 2018; Qian et al., 2021; Li et al., 2021; Ye et al., 2021; Rademakers et al., 2022; Vijayaraman et al., 2019a) reported mean RVD, ranging from 45 to 128 months, and six studies (Shan et al., 2018; Qian et al., 2021; Ye et al., 2021; Rademakers et al., 2022; Vijayaraman et al., 2019a; Yang et al., 2021) provided the success rates and response rates of HPSP upgrading, ranging from 89% to 100% and 62% to 95%, respectively. The follow-up duration ranged from 6 to 36 months. Among the seven studies, five studies were prospective designs (Shan et al., 2018; Qian et al., 2021; Li et al., 2021; Rademakers et al., 2022; Yang et al., 2021) and two were observational studies (Ye et al., 2021; Vijayaraman et al., 2019a). The items of 12 to 15 points in MINORS were used for single-arm studies, which are acceptable for the current meta-analysis (Vijayaraman et al., 2019a; Slim et al., 2003). The details of study characteristics and quality assessment are provided in Tables 1–2.

Table 1 Baseline and procedural characteristics of included studies (Sweeney et al., 2003).

No	Study	Year	Study design	Type	Country	Intervention	Sample size (n)	The diagnosis of PCIM	
1	Vijayaraman et al. (2019a)	2019	Single-arm	Retrospective	USA	HBP	57	A 10% decrease in left ventricular ejection fraction (LVEF) from baseline resulting in LVEF <50% among patients experiencing >20% RV pacing without an alternative cause of cardiomyopathy.	
2	Li et al. (2021)	2021	Single-arm	Perspective	China	LBBP	10	The baseline of the left ventricular ejection fraction (LVEF) is no less than 50%, and the LVEF during follow-up is no more than 40%. The baseline of the LVEF is less than 50%, and the absolute value of the LVEF is reduced by more than 10% during follow-up. The total value of the LVEF is reduced by 10% regardless of the LVEF baseline.	
3	Shan et al. (2018)	2018	Single-arm	Perspective	China	HBP	11	Pacing at least 20% of the time despite device reprogramming to minimize ventricular pacing	
4	Qian et al. (2021)	2021	Control-conhort	Perspective	China	LBBP	13	A ≥10% decrease in left ventricular ejection fraction (LVEF) after RV pacing (ventricular pacing percentage over 20%) with resultant LVEF ≤50% without an alternative cause of cardiomyopathy, and with symptoms of heart failure.	
5	Yang et al. (2021)	2021	Single-arm	Perspective	China	HBP and LBBP	34 (29/5)	(1) Patients with prior RVP implantation (including DDD and VVI pacemaker.) and the percentage of RVP > 40%; (2) A LVEF of ≤40% caused by a new onset LVEF decrease of >10% from baseline, without other identifiable causes.	
6	Ye et al. (2021)	2021	Single-arm	Retrospective	China	LBBP	19	(1) A >10% decrease in left ventricular ejection fraction (LVEF) after chronic RVP resulting in LVEF ≤ 50%. (2) The pacing percentage of RVP was >40%.	
7	Rademakers et al. (2022)	2022	Single-arm	Perspective	The Netherlands	LBBP	20	Patients with RV pacing (ventricular pacing percentage over 70%), the value of the LVEF is reduced by at least 10%, resulting in LVEF <50%.	

Table 2 Baseline and procedural characteristics of included studies (Kiehl et al., 2016).

No.	Study	Year	Age (Year)	Gender/Male (n)	RVD (m)	Follow-up (m)	Response rate (%)	Success rate (%)	Quality assessment	
1	Vijayaraman et al. (2019a)	2019	71.0 ± 13.4	33	78.8 ± 79.4	25 ± 24	95	93	13	
2	Li et al. (2021)	2021	70.8 ± 7.9	5	82.76 ± 45.21	12			12	
3	Shan et al. (2018)	2018	70.6 ± 12.9	9	71 ± 43.6	36.2	62	89	12	
4	Qian et al. (2021)	2021	71.7 ± 9.5	9	128.4 ± 58.8	10.4 ± 6.1	69	93	12	
5	Yang et al. (2021)	2021	69.69 ± 13.8	22		11.52 ± 5.40	90	94	13	
6	Ye et al. (2021)	2021	70.2 ± 8.6	11	76.4 ± 33.5	12	90	95	12	
7	Rademakers et al. (2022)	2022	77 ± 10	14	45.6 ± 55.08	6	80	100	12	

LVEF and NYHA classification

Seven studies reported LVEF. The random-effects model was used for the meta-analysis (I2 = 74%). The results showed that HPSP upgrading significantly improved LVEF in PICM patients (95% CI [11.21–15.61], I2 = 74%). Sensitivity analysis showed that the results were stable.

According to the follow-up time, PICM patients on HPSP upgrading were divided into the group with a follow-up time of ≤6 months and the group with a follow-up time of >6 months. HPSP improved LVEF by 13.20% (95% CI [11.18–15.22], N = 6, I2 = 45%) in the short term (≤6 months), and by 14.25% (95% CI [9.88–18.63], N = 3, I2 = 86%) in the long term (>6 months). There was no significant difference in LVEF at different follow-up time points (P = 0.67) (Fig. 2).

Figure 2 Forest plots of LVEF.

LVEF, left ventricular ejection fraction.

Five studies reported NYHA classification, and the random-effects model was used (I2 = 89%). The results showed that HPSP upgrading improved NYHA classification by 1.02 (95% CI [−1.41 to −0.63], I2 = 89%). According to subgroup analysis by follow-up duration (≤6 months vs >6 months), HPSP improved the NYHA classification by 1.30 (95% CI [−1.68 to −0.92], N = 6, I2 = 74%) in the short term (≤6 months) and by 0.63 in the long term (>6 months) (95% CI [−0.82 to −0.43], N = 3, I2 = 0%). There was a significant difference in NYHA classification between different follow-up time points (P < 0.01) (Fig. 3).

Figure 3 Forest plots of NYHA classification.

QRSd

Six studies reported QRSd, and a meta-analysis was conducted using the fixed-effects model (I2 = 22%). The meta-analysis showed that HPSP upgrading shortened QRSd by 60.85 ms (95% CI [−63.94 to −57.75], I2 = 22%). Sensitivity analysis showed that the results were stable. According to subgroup analysis by follow-up time (≤6 months vs >6 months), HPSP shortened QRSd by 63.74 ms (95% CI [−67.79 to −59.68], N = 2, I2 = 0%) in the short term (≤6 months) and by 56.79 ms (95% CI [−61.59 to 51.98], N = 4, I2 = 0%) in the long term (>6 months). A significant difference was observed in QRSd between different follow-up time points (P = 0.03) (Fig. 4).

Figure 4 Forest plots of QRSd.

QRSd, QRS duration.

Echocardiography parameter

LVESV

Four studies reported LVESV, and a meta-analysis was conducted using the random-effects model (I2 = 76%). The meta-analysis showed a 36.85 ml decrease in LVESV in PICM patients receiving HPSP upgrading (95% CI [−50.49 to −23.21], I2 = 76%). According to subgroup analysis by follow-up time (≤6 months vs >6 months), a 32.46 ml decrease (95% CI [−53.18 to −11.75], N = 2, I2 = 85%) in LVESV was observed in the short term (≤6 months), while a 44.70 ms decrease (95% CI [−58.17 to −31.22], N = 1, I2 = 0%) was found in the long term (>6 months). No significant difference was found in LVESV between different follow-up time points (P = 0.33) (Fig. 5).

Figure 5 Forest plots of LVESV.

LVESV, left ventricular end systolic volume.

LVEDD

Five studies reported LVEDD, and a meta-analysis was conducted using the random-effects model (I2 = 53%). The meta-analysis showed a 5.93 mm decrease in LVEDD in PICM patients on HPSP upgrading (95% CI [−7.68 to −4.19], I2 = 53%). According to subgroup analysis by follow-up time (≤6 months vs >6 months), a 6.04 mm decrease (95% CI [−8.70 to −3.37], N = 2, I2 = 66%) in LVEDD was observed in the short term (≤6 months), while a 5.43 mm decrease (95% CI [−7.34 to −3.53], N = 1, I2 = 0%) was found in the long term (>6 months). No significant difference was found in LVEDD between different follow-up time points (P = 0.72) (Fig. 6).

Figure 6 Forest plots of LVEDD.

LVEDD, left ventricular end-diastolic diameter.

Electrical parameters of the pacemaker

Five studies reported the electrical parameters of the pacemaker. The meta-analysis showed a 0.07 V increase in pacing threshold (95% CI [0.01–0.13], I2 = 78%), a 0.02 mV increase in sensed R-wave amplitude (95% CI [−0.85 to 0.90], I2 = 55%), and a 31.12 Ω decrease in impedance (95% CI [−69.62 to 7.39], I2 = 90%) in PICM patients receiving HPSP upgrading. Subgroup analysis by follow-up time was conducted (≤6 months vs >6 months). The follow-up time ≤6 months was reported in four studies, and the results showed a 0.05 V increase in pacing threshold (95% CI [−0.04 to 0.14], I2 = 55%), a 0.29 mV increase in sensed R-wave amplitude (95% CI [1.38–1.96], I2 = 73%), and a 34.26 Ω decrease in impedance (95% CI [−84.24 to 15.73], I2 = 74%). The follow-up time >6 months was reported in four studies, and the results showed a 0.1 V increase in threshold (95% CI [0.07–0.13], I2 = 0%), a 0.13 mV increase in sensed R-wave amplitude (95% CI [−0.62 to 0.36], I2 = 0%), and a 27.65 Ω decrease in impedance (95% CI [−94.71 to 39.42], I2 = 93%). The random-effects model was used for data analysis if there was heterogeneity between the two subgroups; otherwise, the fixed-effects model was used. The results suggested that the pacing threshold remained stable during follow-up (Figs. 7 and 8) (Fig. S1).

Figure 7 Forest plots of threshold.

Figure 8 Forest plots of sensed R-wave amplitude.

Subgroup analysis based on pacing types

The PICM patients were divided into the HBP group and LBBP group according to different pacing types. The subgroup analysis by pacing types revealed that HBP improved LVEF by 13.28% (95% CI [−11.64 to 14.92], I2 = 36%), reduced the NHYA classification by 1.18 (95% CI [−1.97 to −0.39], I2 = 90%), shortened QRSd by 63.16 ms (95% CI [−67.00 to −59.32], I2 = 17%), lowered LVEDD by 4.12 mm (95% CI [−5.79 to −2.45], I2 = 0%), increased pacing threshold by 0.14 V (95% CI [0.05–0.33], I2 = 49%), elevated sensed R-wave amplitude by 0.20 mV (95% CI [−2.54 to 2.13], I2 = 0%), and reduced impedance by 11.10 Ω (95% CI [−42.18 to 19.98], I2 = 0%). Meanwhile, LBBP improved LVEF by 14.43% (95% CI [−13.01 to 15.85], I2 = 40%), ameliorated NHYA classification by 0.95 (95% CI [−1.33 to −0.58], I2 = 85%), shortened QRSd by 57.98 ms (95% CI [−62.52 to −53.25], I2 = 0%), lowered LVEDD by 6.26 mm (95% CI [−9.87 to −2.64], I2 = 68%), increased pacing threshold by 0.07 V (95% CI [0.00–0.13], I2 = 87%), elevated sensed R-wave amplitude by 0.14 mV (95% CI [−1.22 to 1.49], I2 = 76%), and reduced impedance by 25.14 Ω (95% CI [−82.38 to 32.09], I2 = 93%). The random-effects model was used for data analysis if there was heterogeneity between the two subgroups; otherwise, the fixed-effects model was used. The results are shown in Table 3.

Table 3 The effect size of outcomes after HBP/LBBP upgrading.

Outcomes	Upgrading methods	Effective size (95% CI)	I2	
LVEF	HBP	13.28% (95% CI [−11.64 to 14.92])	I2 = 36%	
	LBBP	14.43% (95% CI [−13.01 to 15.85])	I2 = 40%	
NHYA classification	HBP	−1.18 (95% CI [−1.97 to −0.39])	I2 = 90%	
	LBBP	−0.95 (95% CI [−1.33 to −0.58])	I2 = 85%	
QRSd	HBP	−63.16 (95% CI [−67.00 to −59.32])	I2 = 17%	
	LBBP	−57.98 (95% CI [−62.52 to −53.25])	I2 = 0%	
LVEDD	HBP	−4.12 (95% CI [−5.79 to −2.45])	I2 = 0%	
	LBBP	−6.26 (95% CI [−9.87 to −2.64])	I2 = 68%	
Threshold	HBP	0.14 (95% CI [0.05–0.33])	I2 = 49%	
	LBBP	0.07 (95% CI [0.00–0.13])	I2 = 87%	
Sensed R-wave amplitude	HBP	−0.20 (95% CI [−2.54 to 2.13])	I2 = 0%	
	LBBP	−0.14 (95% CI [−1.22 to 1.49])	I2 = 76%	
Impedance	HBP	−11.10 (95% CI [−42.18 to 19.98])	I2 = 0%	
	LBBP	−25.14 (95% CI [−82.38 to 32.09])	I2 = 93%	
Note:

LVEF, left ventricular ejection fraction; QRSd, QRS duration; LVEDD, left ventricular end-diastolic diameter; HBP, His-bundle pacing; LBBP, left bundle branch pacing.

Discussion

To our knowledge, this study is the first systematic review and meta-analysis evaluating the effectiveness of upgrading to HPSP in PICM patients. The main findings were as follows:(1) Upgrading to HPSP improved LVEF and NYHA classification, shortened QRSd, reversed ventricular structure LVESV and LVEDD, and provided stable electrical parameters in PICM patients. The degree of improvement of QRSd and LVESV increased with a longer follow-up time. (2) The success rate of HPSP upgrading in PICM patients is high and the response is good. The efficacy of HBP is not inferior to LBBP, and there was no significant difference in efficacy between the two modes.

A series of studies have shown that HBP can produce favorable clinical outcomes in patients with CRT indications (Shan et al., 2018; Qu et al., 2021). The HIS-SYNC trial showed that HBP resulted in a shorter QRSd and was equally effective in improving cardiac function, including LVEF and NYHA classification compared to CRT (Upadhyay et al., 2019). Sharma et al. (2018) noted that HBP had a positive clinical response in patients who did not respond to CRT. LBBP, a new pacing method based on HBP, had been proven to effectively achieve left ventricular electromechanical synchrony and improve cardiac function. LBBP is believed to be easier-to-operate and have more stable pacing parameters. Li et al. (2021) compared the effect of LBBP and BVP in a multicenter non-randomized controlled trial. The results showed that LBBP had shortened QRSd, improved LVEF, better echocardiographic response, and lower and stable threshold at 6-month follow-up compared to BVP (Duan et al., 2020). This meta-analysis showed that HBSP improved cardiac function and reversed left ventricular remodeling in PICM patients, which was consistent with the previous studies. This means that HPSP could significantly improve LVEF, NYHA classification, QRSd, LVESV, and LVEDD and produce stable electrical parameters of a pacemaker for PICM patients. In addition, there were more significant decreases in QRSd and LVESV with the extension of follow-up time. This may help explain the need for sufficient time to reverse ventricular remodeling and restore cardiac function after HPSP. The reason why HPSP achieves mechanical reverse remodeling in CRT patients without response is a shorter QRSd. A greater dramatic reduction in QRSd indicates a greater possibility of reversing cardiac remodeling (Khurshid et al., 2014; Kim et al., 2018; Khurshid et al., 2016). CRT could not fully simulate the rapid ventricular activation of His-Purkinje system or achieve a fully standardized QRSd. In addition, our study showed that the success rate of HPSP upgrading was high, which has been proved in 209 patients, 195 of whom were successfully upgraded (93%). During the follow-up time, 133 (86%) patients had a good response.

The effectiveness of HBP vs LBBP in the treatment of PICM remains inconclusive. As a kind of physiological pacing, HBP induces ventricular systolic synchrony through the intrinsic His-Purkinje conduction system. In contrast, LBBP usually can avoid stimulating pathological or disease-susceptible areas, and then generates true conduction system pacing. LBBP had the advantages over HBP in better perception, easier manipulation, and lower long-term thresholds. Additionally, the site of LBBP pacing was close to the block site, which could compensate for an increased threshold of HBP caused by the progression of disease conduction (Huang et al., 2017; Hu et al., 2019). The study by Vijayaraman et al. (2019a) involved 85 patients with long-term right ventricular pacing and found that during follow-up, patients with PICM had improved cardiac function, reversed ventricular remodeling, and no significant disease progression that could not be corrected by HBP. The study by Yang et al. (2021), which involved 34 PICM patients on HPSP (including 29 PICM patients on HBP upgrading and 5 PICM patients with LBBP upgrading), showed significantly shortened QRSd, improved LVEF and NYHA classification, and better capture threshold, and R wave in such patients.

Numerous investigations have demonstrated that HBP upgrading is not restricted by the progression of conduction dysfunction, which is extremely rare, even in PICM patients (Vijayaraman et al., 2019a, 2017, 2015). The success of HBP upgrading depends largely on the experience of operators because bypassing sites of conduction block distal to His bundle or proximal to the left bundle could increase the success rates. Our analysis showed that compared with LBBP, HBP shortened QRSd (P = 0.09). HBP is paced most physiologically through the His-Purkin field conduction system, which is consistent with our definition. There was no significant difference in other outcomes. The results showed that the efficacy of HBP was equivalent to LBBP. Although HPSP, as a more physiological pacing method, has been shown to restore electromechanical ventricular synchrony and improve cardiac function in PICM patients in many trials, most of them were observational designs with a small sample size. Hence, more large-sample studies are needed to verify our findings.

Limitation

This meta-analysis had the following limitations. Firstly, although the sensitivity analysis showed the reliability of results, the heterogeneity of this meta-analysis was high. This may be attributed to different conduction abnormalities at different sites, various right ventricular pacing sites, different operator experiences, and the limited methodological quality of included studies. Secondly, the sample size was small, which may affect the stability of the outcomes, reduce the detection efficiency, and lead to research bias. Thirdly, only seven studies were included and they are not RCTs. Therefore, more high-quality and large-scale studies with longer follow-up duration are needed to verify the above findings.

Conclusion

This meta-analysis showed that HPSP upgrading could significantly improve left ventricular function, reverse ventricular remodeling, and provide stable electrical parameters for PICM patients. The efficacy of HBP is not inferior to LBBP, and they are equivalently effective. More large-scale studies are needed to verify the efficacy and safety of HPSP upgrading in PCIM patients.

Supplemental Information

Supplemental Information 1 PRISMA checklist.

Click here for additional data file.

Supplemental Information 2 Systematic Review and/or Meta-Analysis Rationale.

Click here for additional data file.

Supplemental Information 3 Forest plots of Impedance.

Click here for additional data file.

We would like to thank the researchers and study participants for their contributions.

Additional Information and Declarations

Competing Interests

Author Contributions

Data Availability

The authors declare that they have no competing interests.

Nian Tang conceived and designed the experiments, performed the experiments, prepared figures and/or tables, authored or reviewed drafts of the article, and approved the final draft.

Xiaoxiao Chen conceived and designed the experiments, prepared figures and/or tables, and approved the final draft.

Hongfei Li performed the experiments, authored or reviewed drafts of the article, and approved the final draft.

Denghong Zhang analyzed the data, prepared figures and/or tables, and approved the final draft.

The following information was supplied regarding data availability:

Data sharing not applicable to this article, as no datasets were generated or analysed during the current study.

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
