# Peer review of "Beneficial effects of upgrading to His-Purkinje system pacing in patients with pacing-induced cardiomyopathy: a systematic review and meta-analysis"

_PeerJ, doi:10.7717/peerj.16268_

## Round 0.1 · original submission · Major Revisions

The manuscript requires a number of Major Revisions.

Reviewer 1 ·

Basic reporting

no comment

Experimental design

no comment

Validity of the findings

no comment

Additional comments

1.As shown in Figure 1, the author used the word "other" to provide the reasons for excluding the literature, which is not appropriate. Please provide specific reasons.
2.The color scheme of the images in the manuscript is neither aesthetically pleasing nor conducive to a good reading experience. Please make changes accordingly.
3.I suggest the author make appropriate changes to the introduction as there are a few expressions that may not be idiomatic in English. Additionally, there are some areas where the logic could be improved, although it does not affect the overall comprehension.
4.Conclusion section: L 314 -L 315 is confusing.
5. In Tab 3, not I2, but rather I^2.

Reviewer 2 ·

Basic reporting

This meta-analysis evaluated the on the effectiveness the application of hipkinje system pacing for hipkinje system pacing, and found that HPSP could significantly improve cardiac function, reverse ventricular remodeling and produce stable electrical parameters of pacemakers for PICM patients. This work is well-designed with sufficient clinical value. I have following suggestions:
First, the language should be further revised. Acronyms should be fully spelled when they are first used in abstract and the body text.
Second, some references were too old especially in the introduction part, such as reference 1.
Three, whether the included populations received cardiac resynchronization therapy before received HPSP. Please give a subgroup analysis if possible.
Four, in figure 2, there were 2 “Yang Ye 2021” in the subgroup of follow-up time >6months. Similar problems occurred in Figure 5,6, and 7. Please explain and revise the figures.

Experimental design

the work is well-designed and concucted to the PRISMA guideline and registered in the PROSPERO.

Validity of the findings

First, whether there was publication bias in this meta-analysis. To better evaluate the findings,please give the funnel plot for the main outcome.
Second, the conclusions " The efficacy of HBP upgrading is not inferior to LBBP, both of them are comparable." seems to lack of evidence. The results didn't give a direct comparisons.

Additional comments

none

·

Basic reporting

no comment

Experimental design

no comment

Validity of the findings

no comment

Additional comments

The authors conducted a meta-analysis of previously published literature to study the effectiveness of hipkinje system pacing therapy in patients with PICM, and finally concluded that HPSP can significantly improve cardiac function, reverse ventricular remodeling, and The electrical parameters of the pacemaker were stable. The results of this paper can provide some guidance for the choice of treatment methods for PICM patients. However, there are still some small problems in the article, I hope the author can explain:
1. When searching literature, the author searched several databases such as PubMed, Embase, Web of Science, and the Cochrane Library. Are these databases enough? Are there other databases that can be searched? Does it lead to fewer studies included in the end?
2. What is the reason for the high heterogeneity obtained from Meta analysis? What method did the author use to solve this problem?
3. Which method is used for the sensitivity analysis? It doesn't seem to be clearly stated in the text.
4. Has the paper tested for possible bias such as publication bias?
5. "PICM" in the background should have a full description of the acronym when it appears for the first time.
6. There are some grammatical errors in the full text, please revise.

---

## Round 0.2 · accepted · Accept

The quality of this manuscript has improved after revisions.

Reviewer 1 ·

Basic reporting

no comment

Experimental design

no comment

Validity of the findings

no comment

Additional comments

The manuscript has been revised according to the comments.

Reviewer 2 ·

Basic reporting

no comment

Experimental design

no comment

Validity of the findings

no comment

Additional comments

The quality of this manuscript has improved after revisions.